# A Novel and Automated Approach to Classify Radiation Induced Lung Tissue Damage on CT Scans

**DOI:** 10.3390/cancers14051341

**Published:** 2022-03-05

**Authors:** Adam Szmul, Edward Chandy, Catarina Veiga, Joseph Jacob, Alkisti Stavropoulou, David Landau, Crispin T. Hiley, Jamie R. McClelland

**Affiliations:** 1Centre for Medical Image Computing, Department of Medical Physics and Biomedical Engineering, University College London, London WC1E 6BT, UK; e.chandy@ucl.ac.uk (E.C.); c.veiga@ucl.ac.uk (C.V.); j.jacob@ucl.ac.uk (J.J.); alkisti.stavropoulou.16@ucl.ac.uk (A.S.); j.mcclelland@ucl.ac.uk (J.R.M.); 2Sussex Cancer Centre, Royal Sussex County Hospital, Brighton BN2 5BE, UK; 3UCL Cancer Institute, University College London, London WC1E 6BT, UK; dblandau@gmail.com (D.L.); crispin.hiley@ucl.ac.uk (C.T.H.); 4UCL Respiratory Department, University College London Hospital, London NW1 2PG, UK; 5University College Hospital, University College London, London NW1 2BU, UK

**Keywords:** radiation induced lung damage, lung segmentation, lung tissue classification, deep learning

## Abstract

**Simple Summary:**

Radiation-induced lung damage (RILD) is a common side effect of treating lung cancer with radiotherapy (RT). RILD is visible on CT imaging, and its radiological appearance can vary dramatically from patient to patient as well as across different sub-regions of the lung and treatment volumes. A classification system for RILD able to differentiate radiological damage on a local level would allow us to better understand the underlying patterns of RILD, see how they change over time post irradiation, and link it with clinical outcomes. In this work we propose a five-class morphological lung tissue classification system that can describe parenchymal tissue changes at the voxel level. The classifier was implemented in a fully automated manner using an optimised deep-learning method, then trained and tested using data acquired through a multi-centre clinical trial. The proposed method performed well on an unseen testing dataset. The automated segmentation achieved considerable overlap with manual segmentations (ranging between 26% and 98% for the five classes) and was graded as acceptable by a clinical expert in 88% of cases. This demonstrates it to be suitable for application on a large dataset to help uncover different patterns of changes in the population.

**Abstract:**

Radiation-induced lung damage (RILD) is a common side effect of radiotherapy (RT). The ability to automatically segment, classify, and quantify different types of lung parenchymal change is essential to uncover underlying patterns of RILD and their evolution over time. A RILD dedicated tissue classification system was developed to describe lung parenchymal tissue changes on a voxel-wise level. The classification system was automated for segmentation of five lung tissue classes on computed tomography (CT) scans that described incrementally increasing tissue density, ranging from normal lung (Class 1) to consolidation (Class 5). For ground truth data generation, we employed a two-stage data annotation approach, akin to active learning. Manual segmentation was used to train a stage one auto-segmentation method. These results were manually refined and used to train the stage two auto-segmentation algorithm. The stage two auto-segmentation algorithm was an ensemble of six 2D Unets using different loss functions and numbers of input channels. The development dataset used in this study consisted of 40 cases, each with a pre-radiotherapy, 3-, 6-, 12-, and 24-month follow-up CT scans (*n* = 200 CT scans). The method was assessed on a hold-out test dataset of 6 cases (*n* = 30 CT scans). The global Dice score coefficients (DSC) achieved for each tissue class were: Class (1) 99% and 98%, Class (2) 71% and 44%, Class (3) 56% and 26%, Class (4) 79% and 47%, and Class (5) 96% and 92%, for development and test subsets, respectively. The lowest values for the test subsets were caused by imaging artefacts or reflected subgroups that occurred infrequently and with smaller overall parenchymal volumes. We performed qualitative evaluation on the test dataset presenting manual and auto-segmentation to a blinded independent radiologist to rate them as ‘acceptable’, ‘minor disagreement’ or ‘major disagreement’. The auto-segmentation ratings were similar to the manual segmentation, both having approximately 90% of cases rated as acceptable. The proposed framework for auto-segmentation of different lung tissue classes produces acceptable results in the majority of cases and has the potential to facilitate future large studies of RILD.

## 1. Introduction

Lung diseases are one of the leading causes of death worldwide [1], with lung cancer being the most common cause of cancer death [2]. Radiation-induced lung damage (RILD) is a common side effect of treatment for lung cancer and is one of the main factors reducing quality of life in lung cancer survivors [3]. RILD is usually distinguished into (a) acute phase appearances—pneumonitis, which occurs within 6 months following radiotherapy, and (b) permanent fibrosis, stabilising up to 24 months after the radiotherapy [4]. Until recently, poor long-term survival of lung-cancer patients limited interest in RILD research [5]. Current state-of-the-art treatment, however, results in longer survival [6]. Therefore it is more important to consider the quality of life of survivors and, in turn, the underlying mechanisms of RILD.

RILD can be characterized using computed tomography (CT) imaging. CT scans are routinely acquired prior to radiotherapy treatment to assess the tumour and the lung, and CT imaging is repeated after radiotherapy to monitor for disease recurrence and assess for RILD. The follow-up scans can be used to study lung damage associated with radiotherapy by comparing the imaging to the baseline scans obtained before treatment. Typical post-radiotherapy radiological findings include parenchymal damage, lung volume shrinkage, and anatomical distortion, which can be used to describe and quantify RILD with image analysis techniques [7]. Our team has previously proposed a suite of CT imaging-based RILD biomarkers [8] that describe common changes in the anatomy and shape of the respiratory system. They characterize normal lung volume shrinkage, increase in parenchymal consolidation volumes, and changes in: shape of the lungs, diaphragm, central airways, mediastinum, and pleura. Their applicability has been successfully presented on a cohort of homogeneously treated patients on serial CT imaging of up to 24 months post-RT [9]. Most of the RILD biomarkers were focused on describing and measuring changes to the shape and anatomy of the lungs rather than morphology of lung parenchyma. The original biomarker used to characterise parenchymal change quantified ‘consolidation volume’ as the ratio of high-intensity volume normalised to the contralateral lung. Such a binary classification of parenchymal tissue, based on thresholding, included vessels as part of the volume of damaged lung and over-simplified the complex gradation of changes visually observed in the scans. Furthermore, thresholding is susceptible to acquisition artefacts and intensity variability caused by different imaging protocols (inspiration, expiration, contrast enhancement, 4D CT, etc). An additional challenge might be related to the fact that a systematic increase in volume of the contralateral lung post-RT has been observed [9]. Therefore, that approach might not be well suited for distinguishing morphological sub-types or accounting for changes in the contralateral lung.

The ability to identify and quantify localised RILD changes within the lung parenchyma can provide an additional dimension to the study of RILD. Ultimately, this will allow us to track local disease involvement and longitudinal evolution of damage and relate it to radiotherapy dose and clinical outcomes [10]. Analysis of the temporal evolution of RILD parenchymal changes can provide new insights into radiotherapy dose and time relations. Early and accurate diagnosis of different types of lung parenchymal changes has already been shown to be crucial in ensuring that patients with interstitial lung disease (ILD) are treated optimally [11]. That, however, required detailed consideration of clinical, radiological, and histopathological features, including different types on lung tissue patterns.

The existing global lung tissue damage scoring systems, such as the Radiation Therapy Oncology Group (RTOG) or European Organization for Research and Treatment of Cancer (EORTC), describe radiologic parenchyma changes in the lungs as slight, patchy, or dense [12]. They can be subjective, with users regularly interpreting patchy areas as ground-glass opacities and dense areas as consolidation. In the Common Terminology Criteria for Adverse Events (CTCAE), the degree of “radiologic pulmonary fibrosis” can range from <25% to <75% for grades 1 to 3, and in grade 4 includes the presence of severe “honeycombing” [13]. The differences in RTOG/EORTC and CTCAE guidelines lead to significant variations in grading depending on the system used. For instance, in the multicentre, non-randomized, phase 1/2 chemo-radiation trial of stage II/III non-small cell lung cancer, the IDEAL-CRT [14], 12 months follow-up scans were mostly scored as 2 or 3 using RTOG classification, at the same time being given grade 1 using CTCEA classification [8]. This is because across the majority of scans RILD changes were present (thereby scoring 2 and 3 in RTOG), but were restricted in terms of volume (resulting in grade 1 CTCEA scores). Another limitation of these approaches is their global nature, where a single score is given to the whole scan [8,15]. These scoring criteria are therefore inadequate for detailed descriptions of the complex heterogeneous nature of the RILD parenchymal changes and cannot describe the changes in a localised, voxel-wise manner.

The local parenchymal changes, especially their spatial distribution and temporal evolution due to RILD, have not been widely studied. There are studies looking directly into mean Hounsfield unit (HU) changes as a measure of lung density changes associated with RILD [16,17]. Bernchou et al. investigated regional CT density changes following intensity modulated radiotherapy (IMRT) for non-small-cell lung cancer (NSCLC) with relation to the prescribed local doses [18]. The analysis relies purely on HU as the lung density description, which might be susceptible to the level of inhalation, contralateral lung hyperinflation, imaging artefacts, or acquisition protocols, and does not incorporate texture features of the lung parenchyma. There have been attempts to classify and quantify RILD using multiple radiomics-based approaches [19], where 20 features were identified in randomly chosen patches to assess the correlation between change in the features before and after radiotherapy with relation to the prescribed dose. In that study, most features were strongly related to the mean HU of the patch and only higher order features represented patterns. Another study looked at differences in inter- and intra-observer variability in delineation of fibrotic lung regions [20]. However, there was no comprehensive classification method introduced dedicated to studying the general morphology of RILD. In a recent study, Al Feghali et al. looked at lung density changes relying directly on differences in HUs of CT scans after performing rigid registration between different time point images [21]. Such an approach is prone to errors originating from different levels of inhalation that rigid registration cannot compensate for, requires CT scans acquired with the same acquisition protocol (without contrast, and the same CT reconstruction kernels), and limits the interpretability of observed lung tissue patterns. In [22] the authors highlight that RILD could be divided into even finer temporal stages: early, latent, exudative, intermediate, and fibrotic phases. However, the lack of a tissue classification system suitable for describing the local changes present at different stages, along with the need for manual annotations for such a process, limits the ability to explore their radiological appearances and their evolution over time on a larger dataset. Therefore, the existing approaches either lack ability to describe local radiological changes or are limited in their power to describe the range of morphological patterns encountered in RILD.

There is currently no established classification system of local parenchymal changes due to RILD and no available annotated RILD dataset, which could be used for training an automatic segmentation method. That is most likely for two reasons: first, the definition of lung tissue patterns to be annotated is a challenge itself, and second, performing manual annotations is a laborious task. In the context of other lung diseases, commercial software tools, like CALIPER [23], are available for automatic lung tissue classification. However, CALIPER was designed for automatic segmentation of different radiological lung patterns most commonly observed with interstitial lung disease (ILD) and is not suitable for RILD application. CALIPER requires high spatial resolution scans acquired at breath-hold and works optimally on data acquired with specific reconstruction filters, and it is not intended for contrast enhanced scans, which are commonly used in lung cancer surveillance scans. Existing voxel-wise annotated datasets, which include an ILD dataset [24], are not designed to allow the analysis of RILD, as several patterns included in these datasets are rarely if ever seen in RILD. The other limiting factor of the ILD dataset is its sparse annotation, where only sections of an individual slice, and a limited number of slices from a volumetric scan, were labelled. These labelled areas usually represent regions where the annotator had high confidence in the labels. Such an approach provides good examples for certain tissue classes, but makes global evaluation challenging, particularly for the less represented classes.

The existing methods of describing RILD parenchymal changes are either global in nature and lacked spatial specificity [12,13,15] or do not characterise the different morphological patterns present in the scans in detail [16,17,20], therefore, are not well suited for investigating longitudinal characteristics of RILD parenchymal change. The main goal of this work was to develop, implement, and validate a novel image analysis method for lung tissue classification in the context of RILD. Therefore, our contributions in this work are:The use of a novel ‘two-stage’ data labelling approach that allows us to co-develop:-A novel image-based grading system for RILD that is able to describe at the voxel level the morphological patterns of RILD;-An automated method based on standard deep-learning approaches to perform the classifications;Quantitative and qualitative validation of the automated classification system.

To the best of our knowledge, it is the first attempt to produce an automated, detailed, and voxel-wise description of RILD.

## 2. Materials and Methods

### 2.1. Patient and Imaging Characteristics

Our dataset consisted of 46 patients for which imaging data were available at five time points: before radiotherapy treatment, and at four follow-up time points: 3, 6, 12, and 24 months post-RT. The pre-radiotherapy baseline scan was for most cases a diagnostic CT; a non-diagnostic image, which may be planning (3-dimensional free-breathing or average 4DCT) or PET/CT (breath-hold), was used when a diagnostic scan was not available. These patients were a subgroup of the IDEAL-CRT [14], a nonrandomized phase 1/2 multicentre trial, for which imaging data were available at all five time points. This trial enrolled patients with stage II/III non-small cell lung cancer to receive isotoxic tumor RT doses between 63 and 73 Gy in 30 fractions over 6 weeks (daily) or 63 to 71 Gy in 30 fractions over 5 weeks (2 fractions delivered in a single day once a week) concurrent with two cycles of cisplatin and vinorelbine. Tumor prescription dose was defined to achieve a mean lung dose of 18.2 Gy (in equivalent dose of 2-Gy fractions). Acquisition parameters and image resolution varied both intra- and inter-patient, with most of the scans being diagnostic and acquired at breath-hold. The axial resolution of the scans ranged from 0.57 × 0.57 ${\mathrm{mm}}^{2}$ to 1.40 × 1.40 ${\mathrm{mm}}^{2}$, and slice thickness was in a range of 0.7 mm to 5 mm, amongst all scans across different time points.

### 2.2. RILD Tissue Classification System

As there is uncertainty in the clinical interpretation of the RILD related parenchymal changes seen in CT scans, we decided to propose a set of tissue classes based purely on the morphological appearance of the tissue. The proposed lung parenchyma labels include:**Class** **1:**represents normal, healthy, or emphysematous lung without any high-density abnormality and represents most of the lung parenchymal tissue prior to radiation, as well as areas not affected during the radiotherapy.**Class** **2:**represents areas mostly characterized by changes similar to ground-glass opacity, which is: “hazy increased opacity of lung, with preservation of bronchial and vascular margins. Ground-glass opacity is less opaque than consolidation, in which bronchovascular margins are obscured” [25].**Class** **3:**represents areas with mixed ground-glass opacity and overlaid reticulation.**Class** **4:**represents mostly solid lung tissue, either aerated opaque tissue or tissue with a density just below dense opacity.**Class** **5:**represents homogeneous, dense lung tissue and could represent a number of pathological entities including tumour, pleural effusion, or collapse [26].

Figure 1 shows example images representing the classes described above with corresponding labels in colors, and their distribution in our dataset is shown in Table 1.

The data were manually annotated by an expert (EC, radiation oncologist) in a voxel-wise manner using ITK-SNAP [27]. The initial five cases (across all time points: 25 CT volumes) were reviewed by JJ (thoracic radiologist with 10+ years of experience), who served as an external radiologist. The remaining 35 cases were manually annotated according to the same criteria by EC to create the stage one ground truth labels.

### 2.3. Automated Segmentation Method

As a backbone segmentation method we have applied the well known 2D UNet, which was explicitly designed for medical image segmentation tasks [28]. Only lung regions were presented to the networks; anatomy beyond the lungs was masked. We used manual annotations of the lung tissue to extract lung masks. The input images had a large variety of resolutions and represented slices of varying thickness. The network was presented with images cropped to the lungs and fitted to the size of a fixed input as far as was possible. We kept the original resolution wherever possible, adjusting the cropping range, as that the network was robust to different resolution and scan quality. To avoid downsampling and changing the ratio of the images, we applied a rectangular input images of (288, 384), which resulted in downsampling of only  10% of images. We decided to downsample some of the images rather than use a patch-based approach to allow for inclusion of lung shape knowledge, which might have been lost with patches. The chosen size of input images was made to best fit the lungs from an axial view, which creates a rectangular bounding box, and to allow for multiple reduction stages by the networks at the encoding stage. We applied four reduction stages in the UNet architecture with instance normalisation [29]. Our networks were configured as multiclass segmentation, so the final layer was softmax. We did not apply any form of postprocessing of the output segmentations apart from excluding regions outside the lung using the existing lung masks. Detailed description of the applied network, configurations and parameters can be found in the Appendix A.

Our dataset (46 cases) was split into 40 cases used for development (200 CTs) and 6 cases (30 CTs) which were withheld as a testing dataset. The development dataset was further divided into training (28 cases, 140 CTs) and validation sets (12 cases, 60 CTs). To prevent information leakage, the data were split on a patient basis, i.e., scans from all time points from a single patient went to the same training, validation, or testing subset. The testing dataset was not used during the development stage and no changes to the method were applied after the evaluation on that dataset. The testing dataset was entirely manually annotated and was not included in the two stage annotation approach described above. The test cases were the last to be annotated, only after the labels on training dataset were finalised.

The distribution of the abnormalities in the lungs was uneven (Table 1). Parenchymal change characteristic of RILD is a local effect, and therefore expected to be more pronounced in the regions of the lungs receiving higher radiation doses. This results in a severe label imbalance problem for the segmentation task, where on average approximately 93.7% of total lung volume was assigned with Class 1, and only approximately 6% of total volume was distributed among the other four classes. To mitigate for such a strong class imbalance, only slices where at least 1% of lung tissue was labelled with Classes 2 to 5 were used. This improved the distribution balance, with approximately 85% of the remaining lung volume being labelled with Class 1, and now approximately 15% was distributed among other classes. Using all slices, our dataset consisted of 31,264 2D slices, whereas the dataset containing of at least 1% consisted of 11,237 2D slices.

### 2.4. Two-Stage Ground Truth Data Generation

We applied a two stage ground truth data generation approach, akin to active learning methods [30,31], which is shown schematically in Figure 2. First, the initial ’draft’ manual labels were drawn. We assumed that the labels were not final and would require additional revision or refinement. The ’draft’ labels were then used to train CNN models to perform automated annotations. At the second stage of data annotation, for each of the scans, two segmentations (manual and automatic) were presented to the annotator. The labels generated by the CNNs were manually reviewed and carefully compared with manual labels looking at any discrepancies between them, deciding whether the CNN label or manual label was correct. This required also checking adjacent slices (to detect artefacts that the CNNs may have mislabelled). After that, the labelling (manual or automatic) that was considered to require less correction was chosen to be refined to create the stage two ground truth label. That process was performed for each CT scan. The reason for using such an approach was related to the continuous spectrum of patterns, which implies that there is a large degree of uncertainty when assigning a specific class to some regions. Using the two stage approach we were able to address this subjective and somewhat ill defined problem, as the classes were refined at the same time as performing the labelling. Our chosen joint-machine-and-human learning approach should lead to more consistent labels that better represent the variability in the data than is achievable with a single stage approach.

#### 2.4.1. Stage One

We used an ensemble of three networks in the form of 2D UNets with different loss functions (weighted cross entropy (WCE) or Lovasz loss [32]) and number of filters used (32 or 64). We hypothesized that an ensemble of the networks will give better results than any of the individual configurations, but to confirm this we assessed the individual networks as well as the ensemble. The WCE was adjusted for the frequency of the classes’ appearance, setting its weight depending on the classes presence as a form of mitigation for the class imbalance. The networks with WCE loss was used with 32 filters, whereas Lovasz loss was used with 32 and 64, resulting in three different configurations in total. The results of the individual networks were combined into an ensemble by summing the logit scores of all of the networks in a channel-wise manner and only then applying argmax operation. Such an approach allowed us to take into account the confidence of each of the networks compared to naive averaging of the final labels.

#### 2.4.2. Stage Two

The revised dataset after stage two ground truth data generation was used for training the new ensemble of CNNs that were applied for the classification of subsequent datasets. Our observation from the initial experiments was that the ensemble of the networks performed better than any individual network, therefore we decided to include more networks in our ensemble. For stage two auto-segmentation we used three different loss functions: Dice, Lovasz, and weighted cross entropy. Each of them was employed with 32 and 64 channels at the initial layer. The decision of inclusion of Dice loss was made based on observed variability in the results between Lovasz and CE. There was some level of disagreement between the two and we expected that by enlarging the ensemble of networks we would improve the consistency of the results. We kept the same approach in combining the stage two results in the ensemble by summing the logit scores of all of the networks in a channel-wise manner and only then applying argmax operation.

### 2.5. Evaluation

The quantitative evaluation was performed based on a Dice score coefficient. The Dice score coefficient was calculated for the whole dataset for each of the labels, rather than for each individual scan/patient and then averaged. In such a way we had just one score for each of the labels rather than a mean value and standard deviation. The advantage of such an approach is that it takes into account the fact that in some scans the less represented labels might consist of a very small number of voxels. If in such cases those few voxels are misclassified, it could potentially lead to distorted results. We will refer to the chosen approach of presenting the Dice score coefficient as global Dice.

#### 2.5.1. Stage One/Stage Two Development Dataset Evaluation

The proposed method was first evaluated quantitatively (based on global Dice) on the development dataset using the stage one network configuration and splitting the data into training and validation (equivalent to using a single fold).

After revising the labels and creating the stage two dataset, we trained new networks using the stage two configurations. We evaluated the performance of the stage two networks using the same training and validation split as in the stage one and using global Dice as a metric.

#### 2.5.2. 5-Fold Cross Validation on Development Dataset

Next we performed the quantitative evaluation on the development dataset in a 5-fold cross validation fashion. We used different folds from those used in the stage one experiments. Five experiments were performed with each fold being used as a validation subset and the remaining used for training.

Following that, we retrained the stage two networks on the whole development dataset and performed quantitative and qualitative evaluation on the withheld testing dataset.

#### 2.5.3. Evaluation of the Method on the Test Dataset

Following the quantitative evaluation of the method on the development dataset, we conducted experiments on the hold-out testing dataset of 6 cases (30 CT scans) using global Dice as a metric.

The qualitative evaluation of our method was conducted by presenting the results of manual and automatic segmentation to an independent expert radiologist (JJ). Although JJ was involved in the initial development of the annotations and revised a number of early cases from the development dataset, he had not previously seen the testing cases. The manual and automatic segmentations were presented in a blinded way, randomly renaming them as Seg_A and Seg_B. For each of the cases, JJ evaluated both segmentation results, providing a set of scores between 1 and 3: 1—Acceptable, 2—Minor disagreement, 3—Major disagreement for each of the classes for each of the scans.

## 3. Results

### 3.1. Stage One/Stage Two Development Dataset Evaluation

We present here the numerical results of global Dice (in %) from the development dataset for the networks used for the two stage ground truth labels generation (weighted cross entropy loss—WCE, Lovasz loss—LV, 32 filters at first layer—32, 64 filters at first layer—64), on the draft labels and after the refinement.

In (a) in Table 2, we present the results of the networks’ performance when the draft labels (i.e., before the label refinement approach was applied) were used at stage one. We observed a lower performance on the validation dataset than on the training. The ensemble of the three networks performed better on the validation dataset than any individual network, when means of all classes were considered.

In Table 3, we show a percentage of manual and automatic segmentations after stage one of labelling from the training and validation datasets chosen to be refined in the second stage of ground truth data generation. These represent the segmentations that were considered as closer to the real ground truth, therefore requiring fewer corrections to be made. We show how it differed between the training dataset (seen by the network) and the validation dataset (not seen by the network). A slightly higher number of manual scans were chosen in the validation (18.33%) than in training dataset (12.86%).

Table 2 part (b) summarises the results of the experiment where the stage two networks were retrained on the refined dataset after the proposed two stage ground truth data generation procedure. The global Dice scores in the training subset were comparable with those from (a) in Table 2; only Class 3 showed noticeably lower performance, 53.37% during the initial experiment compared with 39.99% after the refinement. However, the scores in the validation subset improved for all labels, with the largest improvement for the Class 3, from 13.1% to 32.91%. That was just slightly lower than in the training set, 39.99%, compared to 32.91% for the validation dataset. In the initial split, the performance of the method in the validation subset was almost four times lower (13.1% compared to 53.37%), indicating that agreement between the human and the network has improved after the labels had been revised. Although there was still considerable disagreement for Class 3, when images and labels were inspected, we concluded that this was mostly due to ambiguities between classes 2, 3, and 4, and most of the misclassifications were by only one class.

### 3.2. 5-Fold Cross Validation on Development Dataset

After the initial stage one/stage two evaluation, the stage two networks were retrained on the dataset split into different folds. In Table 4, we summarize the results for the 5-fold cross validation, presenting the global Dice scores calculated for each of the classes for each individual networks as well as for their ensembles. The ensemble of all networks gave the best overall results. Even when individual networks or other ensembles gave marginally better results for one class, the scores for some of the remaining classes were lower than the ensemble of all networks. That observations supports our decision to enhance the range of networks used in the ensemble.

A box plot presenting the results for the 5-fold cross validation is shown in Figure 3. It can be observed that results for the validation subsets were performing slightly worse than for the training, with the lowest scores observed for the Class 3, which was also the least represented class. In Table 5, we show numerical details of the 5-fold cross validation, with the mean values for each of the classes, their standard deviations, and confidence intervals.

In Figure 4, we present visual comparison of the segmentations for two cases (two slices for each) across all folds, and in the final row when all development data were used for training. There was little visual difference between the folds or when all folds (whole development dataset) were used for training. For both presented cases (and also overall across the whole dataset) the results from the different folds as well as using all folds are very similar between them and with the ground truth. That indicates that the network can robustly model the tissue classifications for this patient. Subtle differences can be observed between adjacent classes, e.g., 3 and 4, or 4 and 5. For areas where they do not agree, discrepancies are usually only by 1 class, and often the predicted labels could be considered reasonable based on the CT images. No clearly wrong annotations can be identified.

### 3.3. Evaluation of the Method on the Test Dataset

In Figure 5, we show visual results for each of the six test patients as a comparison between the automatic and manual annotations at the time points where the most abnormalities were observed. The visual results from Figure 5 followed a similar pattern to those from Figure 4, with overall good agreement between predicted and ground truth labels.

The discrepancies are again observed mostly between adjacent classes (for instance Class 2 and Class 3). The automatic method seems to label more lung regions as Class 2 than the manual annotator. That observation was later confirmed by a confusion matrix (Figure 6), showing that Class 1 in both datasets was mislabelled in 2% of cases as Class 2. It also can be noted that the automatic methods found it challenging to identify Class 3, mostly confusing it with Class 2. Clearly wrong annotations were mostly observed in the regions with imaging or motion artefacts.

#### 3.3.1. Quantitative Evaluation

For quantitative evaluation we calculated global Dice scores for the whole testing dataset when the whole development dataset was used for training. Table 6 summarises the scores observed for the training dataset and testing dataset (six cases). We observed similar results for the training subset to those reported in Section 3.2. However, the results for the testing dataset were lower than those for the validation from Section 3.2. To investigate where the errors originated from, we calculated confusion matrices from both sets, shown in Figure 6. It can be seen that the errors mostly happened among the adjacent classes, which confirmed our observations from visual inspection of the results conducted earlier. Class 3 was particularly challenging in annotation, where most of the voxels had been incorrectly classified as Class 2, and 23% were classified as Class 1, which is more than were correctly classified (19%).

#### 3.3.2. Qualitative Evaluation

The results of the qualitative scores given by the independent observer for manual and automatic segmentations are summarised in Table 7. Both approaches showed similar levels of acceptable segmentations across all classes, with the manual scoring on average 92.7% compared to 88% for the automatic one, across all classes. Surprisingly, the lowest scores for both methods were given to Class 1, and the highest scores to Classes 3 and 4, which was the opposite to what has been observed in the quantitative evaluation. The main difference between the two annotation methods is that automatic segmentations were identified in approximately 7.3% as requiring major changes, whereas for manual annotations it was reported in only 1.3% of results. The poorer performance of the automatic method was mostly influenced by Classes 1 and 2, where in 13.3% of cases score 3 was given. After reviewing the visual results and based on the notes from the independent observer, we found that most disagreements for these classes were related to mislabelling resulting from imaging artefacts, e.g., blurring of the diaphragm due to breathing motion being labelled as Class 2. The manual and automated annotations had a similar number of minor disagreements, with 6% and 4.7%, respectively. For the manual annotations, more of the minor disagreements were seen in Classes 1 and 2, whereas for the automatic annotations, the minor disagreements were more evenly spread over all classes.

## 4. Discussion

In this work we introduced a novel RILD-dedicated morphological lung tissue classification system. We used a two-stage ground truth label generation method, similar to the active learning approach. We developed a deep-learning framework, involving an ensemble of different 2D methods, to automatically generate the proposed labels. The work presented in this study addresses two challenges, first to introduce a labelling system suitable for capturing changes on longitudinal CT imaging that may be applicable to local RILD parenchymal change, and second to develop an automatic tool for their segmentation from unseen new data. It has been shown before that the global RILD characteristics change longitudinally [9], however, local evolution of lung parenchymal changes remains rarely investigated.

During the development of the lung tissue labels, the main aim was to best capture RILD lung tissue parenchymal patterns in terms of lung tissue density and texture. The labels were devised with close discussion with an experienced thoracic radiologist. The main reason for creating morphological classes, rather than pathophysiological classes (for instance following classes used to describe ILD patterns, as in [23,24]), is that the pathophysiology of RILD parenchymal changes is not yet well understood, so defining classes based on morphology rather than pathophysiology allows for unbiased investigation of the radiologically observed changes in the parenchymal tissue. As RILD is very complicated and influenced by many factors including treatment, genetics, and underlying conditions [22], the existing pre-assumptions originating from other lung pathologies could result in false interpretations. Solely morphological patterns could allow for novel insights in the analysis of their spatial and temporal evolution, without the context (e.g. patient history, treatment, or information from previous scans) that could potentially bias segmentation decisions. Our aim was first to establish a method of measuring the changes that can be observed in the scans. The images were annotated independently at every time point, which limited the bias of pre-assumptions imposed from the previous time points.

After thorough review of our data, we opted for five classes, as this allowed the annotators to robustly and confidently assign distinct labels to the classes, at the same time allowed for their meaningful gradation. In the process of developing the proposed classification system, some other classes were initially considered but eventually combined with one of the proposed classes if they were rarely present or were not well suited to describe RILD changes. For instance, pleural abnormalities were initially considered as a separate class, but this was subsumed under the Class 5 (describing opaque patterns) in order to maintain a purely morphological taxonomy. Pleural thickening is often continuous with lung parenchyma and a definitive boundary can often not be reliably distinguished on CT. Another class that was initially trialled aimed to describe ‘fibrosis’ and included honeycombing and reticulation (both representing irreversible lung damage in fibrotic lung disease). However, it was difficult to distinguish this pattern from traction bronchiectasis occurring on a background of emphysema based on its radiological appearance without context [33], so it was included in Class 3. Additionally, it was a very rare label (present only in a handful of scans with a very small volume), and we assumed that it would be challenging to reliably train and evaluate an algorithm to segment such a minor class. In most of the cases the patients from our dataset presented with pre-existing damage in the lungs, which is unsurprising in a lung cancer population where smoking-induced lung damage is frequent. The proposed labelling system is not only capable of describing the RILD changes, but could potentially also be suitable for measuring other non-radiation induced pathologies, such as pneumonitis in patients receiving immunotherapy rather than radiotherapy. It is still possible to further subdivide the proposed classes, for instance to allow emphysematous lung or lung with air-trapping to be distinguished from normal lung in Class 1. We intend to do that in future work by relabelling emphysematous regions by a network trained on another dataset, for instance, ILD data.

Our method goes far beyond the other approaches investigated so far for RILD parenchymal changes, where only HU changes were considered [16,17]. The proposed method explores a wider range of tissue classes than just fibrosis [20], allowing for a more detailed description of RILD paranchymal changes. Recently, much attention has been focused on COVID-19 lung related pathologies, when the majority of the approaches used binary abnormality classification [34], with only a few studies extending analysis to more classes [35].

The manual labelling process was inherently challenging due to the continuous nature of lung tissue changes, the subjective nature of assigning the labels, and the laborious nature of manual annotation in a voxel-wise manner of each individual scan. CNNs, with their ability to uncover differences in the images, as well as being trained on all of the images at once, had the potential to label the data in a more self-consistent manner. Ultimately, we wanted to develop an automatic CNN-based method to annotate the labels; therefore, we trained CNNs on the stage one manual labels with the aim of using the CNN labels and stage one manual labels to generate a revised set of ’ground-truth’ labels on the development dataset. During the revision step of the stage one manual and CNN generated labels for the same scans, it was often found that the CNN results were more consistent across the dataset. That was most likely due to the fact that the CNNs were effectively labelling them all at once. Indeed, the reason for using the two-stage approach was primarily because we hypothesized that it would result in more consistent and objective labels than doing a single stage of manual labelling.

In the ideal fully supervised approach, the labelling process would be conducted by a group of experts, providing an independent set of labels or by reaching consensus on the labels, which would serve as the ultimate ground truth. In the real world, this is very challenging due to human resource requirements. Based on our experience dealing with such a challenging segmentation task, we would still recommend a two-stage approach to help refine the classes, even if labels from multiple observers were available. The method can be perceived as joint work between a human and a machine, one supervising another, and to a certain extent serving the purpose of two annotators. The proposed two-stage data generation method can be applied to other tasks, where manual ground truth data are required but need to be generated, especially when the labels are initially not clear and subjective. The primary application of such approach would be to labels that are morphological, as CNNs are very well suited for finding underlying similarity in patterns and appearance, where context may act as a confounding factor.

When the final network was evaluated on the hold out test dataset, the global Dice values were noticeably reduced when compared to the values for the validation data. The main reason for this is likely that the two-stage data labelling approach was used to produce the ground truth labels for the development dataset, but not for the test dataset. We did not want to apply the two-stage approach to the test data, as that could have potentially biased the development of the method and hence the results. However, this means that the test labels are likely less consistent and contain higher uncertainty than the labels for the development dataset. When the results of our automated segmentation method were compared visually with the ground truth, such as in Figure 5, they showed a good level of visual agreement, even with very complicated underlying pathology.

The qualitative evaluation supports the conclusion that our method is performing well, with almost as many cases being considered acceptable (88%) as for the manual segmentations (92.7%). We found that although the quantitative results for different classes varied on the testing dataset, ranging from 26% to 98% in terms of global Dice, they all performed more comparably in qualitative evaluation. An interesting observation is that the more represented classes, with theoretically higher confidence in annotation, received lower scores than underrepresented and more uncertain ones. This is in contrast to the quantitative evaluation observations, where more represented classes had superior global Dice values to the underrepresented ones. One potential explanation is that for classes with higher confidence in annotations, it is easier to identify when the segmentation is wrong (or not acceptable) but for the classes where there is more uncertainty it is harder to say the segmentations are definitely wrong, so they are considered acceptable.

The global Dice scores for the different classes were strongly influenced by the prevalence of each class in the data. The classes that included many voxels and were present in most or all slices had high global Dice values, whereas the classes that were only present in a few slices and were completely absent for some scans had low global Dice values [36]. This is a well-known limitation of the Dice metric [37] and is particularly evident for our application where there is also a high degree of uncertainty in the precise voxel-wise labelling of the classes, especially in the hold out test dataset. The Dice metric can be a useful tool for comparing the performance of different networks or methods on the same structures/classes, but it should be used with caution when comparing the performance of a method on different structures/classes, and in general is not an appropriate measure for validating that results are suitable for a specific application.

The data used in our study were from a multi-centre clinical trial. There were therefore significant differences in the scan resolution, acquisition protocols, and application of contrast. Acquisition parameters and image resolution in our dataset varied between patients and between time points for individual patients, with slice thickness ranging from 0.95 to 5 mm. We consider using such diverse data as a major strength of our work, as they better represent the diversity in scans seen in standard clinical practice. Therefore we would expect better generalisation of our trained networks on other datasets, although this will need to be verified and will be the focus of our future work. The diversity in the acquisition protocols and data quality was one of the reasons that supported our decision to apply a 2D approach. Otherwise we would need to resample the data or use patches containing a different anatomical region of the lungs. Using a 2D approach, our method could operate on 2D images similar to how a human observer would look at and segment CT images. That, however, restricted the field of view presented to the networks to just one slice.

One of the identified limitations of our automatic segmentation approach was its susceptibility to breathing motion artefacts or partial volume effect consequences, especially in the scans with larger slice thickness, when healthy regions close to diaphragm were occasionally classified as damaged tissue. These should have been limited at the image acquisition stage, when the scans are acquired at a breath-hold, therefore we were not initially considering any mitigation actions against them. Such artefacts might need an additional preprocessing or assessment step, which would identify cases that we suspect might not be suitable for being automatically processed or would require visual inspection. Alternatively, regions where motion artefacts were identified could be excluded from further analysis or classified as a new ’tissue class’ and used in the automated tissue training stage. Another possibility would be to include a dedicated data augmentation strategy at the training stage to help to limit their influence. The way in which a human observer deals with these challenges is by looking at adjacent slices and determining whether a pattern represents a manifestation of pathology or an acquisition artefact. That could be addressed by either incorporating 3D patches [38,39], which, however, comes with aforementioned challenges, or using only a few adjacent slices to predict labels for the middle slice [40,41], which we plan to explore in future.

In the proposed method we used manually segmented lung masks which were available for our dataset. Accurate segmentation of severely damaged lungs is very challenging by itself, and in this work we wanted to focus on the tissue classification task. For future work we would like to combine the proposed framework with an automatic lung segmentation method so it is suitable for fully automatic analysis of large volumes of data. A combination of the lung segmentation and tissue classification methods would have the ability to identify severely damaged or collapsed lungs, as, for instance, those with high level of opacity. That could help to identify cases where lung segmentation might perform sub-optimally and require manual inspection or corrections. It has been shown in a number of earlier studies that automatic lung segmentation methods tend to perform well in mild to moderately damaged lungs but mostly fail in severely damaged cases [42,43].

In this work we used an ensemble of relatively standard 2D UNets with well-known loss functions, as these have shown promising results in a wide range of medical imaging applications. Although these produced satisfactory results given the challenging and subjective nature of assigning voxel-wise tissue classifications, future work will explore state-of-the-art networks and methods that may give superior results to the standard networks used here.

In future work we aim to use our classes to investigate if and how tissue changes are linked to RILD pathophysiology. We have already conducted an analysis where we applied the presented classification method to investigate the degree of radiological change [44]. In that study, the longitudinal data of 24 months follow up were registered to planning scans using a dedicated multi-channel deformable registration method [45], tailored to deal with large anatomical changes. The analysis was conducted to investigate the distribution and evolution of the lung tissue classes with respect to the dose delivered to the tumour and the change in lung function of lung cancer patients. We observed a strong dose-dependent relationship between the proposed classes characteristics and locally prescribed doses.

## 5. Conclusions

We have proposed and automated a lung tissue classification system capable of describing local RILD parenchymal tissue changes. The method was trained and tested on multi-centre clinical trial data, proving to be suitable for application on larger datasets to help uncover different patterns of changes in the population. The auto-segmentation ratings were similar to the manual segmentations, having 88% of cases rated as acceptable by an external radiologist evaluation the scores. The results indicate that the developed method is of sufficient quality to be used for future studies of RILD.

## Figures and Tables

**Figure 1 cancers-14-01341-f001:**
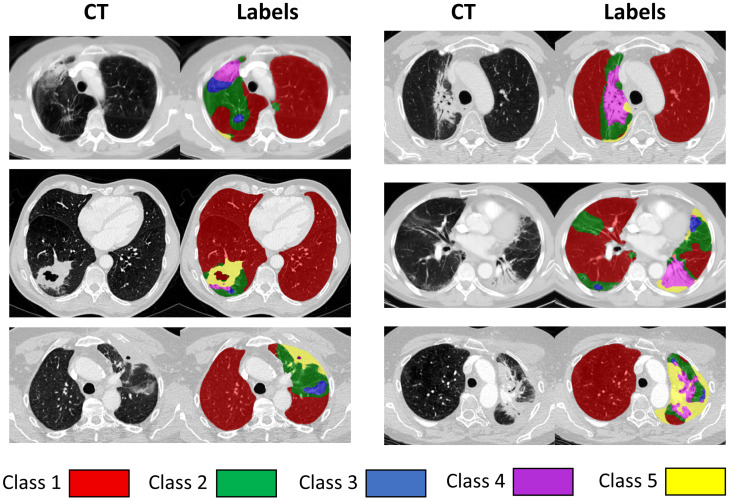
Examples of scans representing all classes and corresponding label maps.

**Figure 2 cancers-14-01341-f002:**
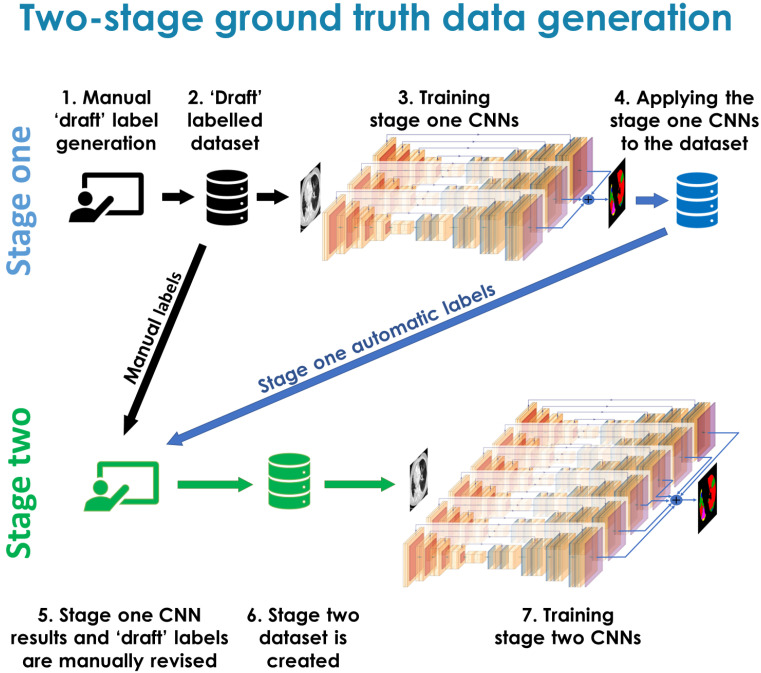
Workflow of the proposed two stage ground truth data generation approach.

**Figure 3 cancers-14-01341-f003:**
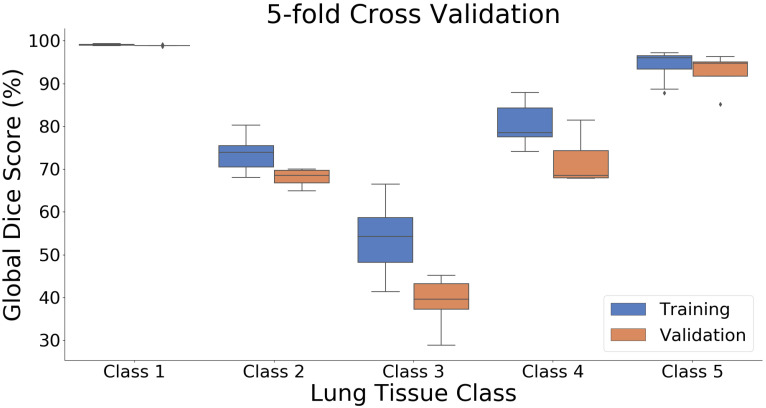
Box plot of global Dice (in %) scores for 5-fold cross validation on the development dataset for five tissue classes. The global Dice distribution is shown with respect to the folds.

**Figure 4 cancers-14-01341-f004:**
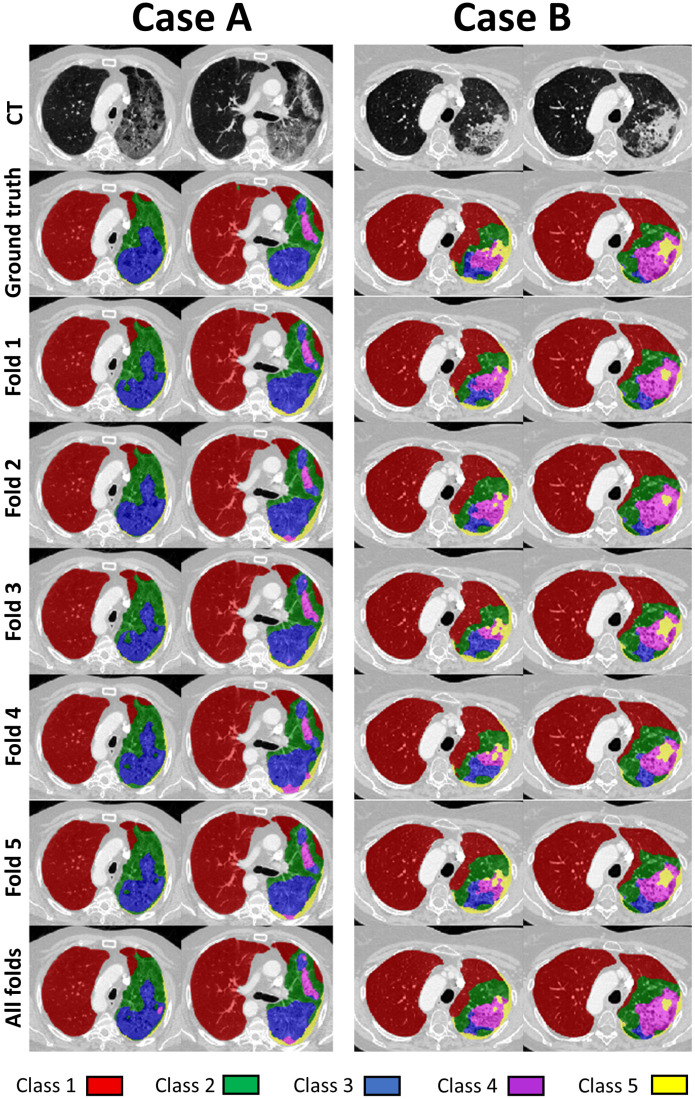
Example of comparison between the folds for two cases (A and B). Overall reasonably good agreement between predicted and ground truth labels can be observed. For areas where they do not agree, discrepancies are usually only by 1 class, and often the predicted labels could be considered reasonable based on the CT images.

**Figure 5 cancers-14-01341-f005:**
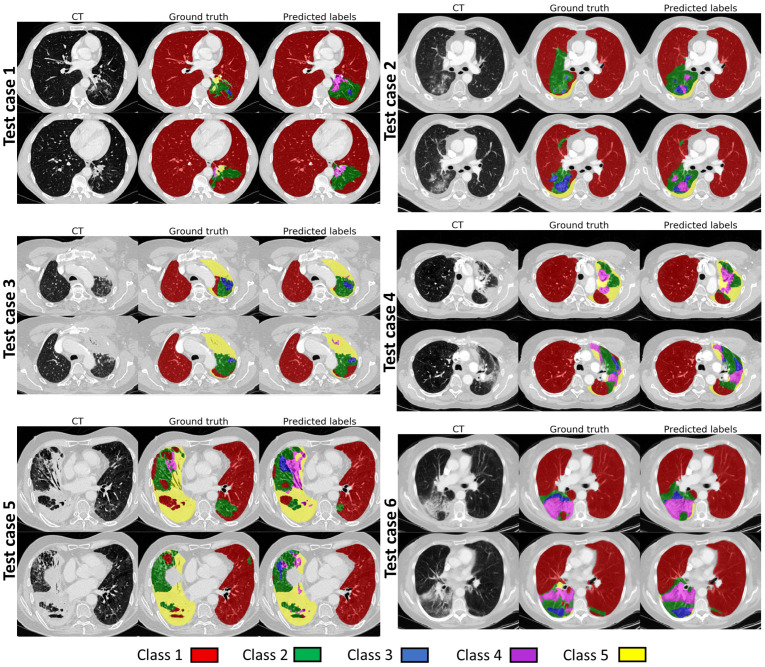
Visual comparison of the segmentation results on the testing set. For each of the cases the time point where the most of the lung parenchymal abnormality was present was chosen.

**Figure 6 cancers-14-01341-f006:**
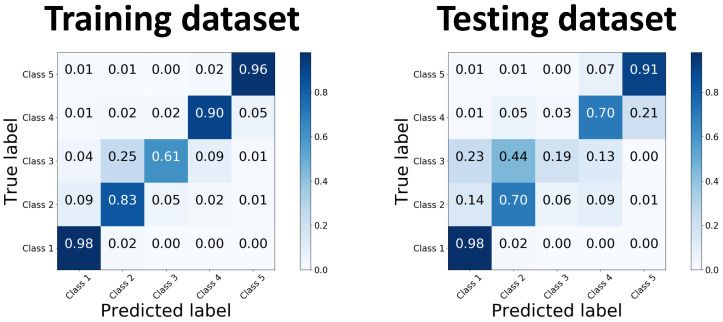
Comparison of confusion matrices calculated for training and testing datasets.

**Table 1 cancers-14-01341-t001:** The distribution of the label classes in the development dataset when all slices were used for training and when only slices where at least 1% of pathology (Classes 2 to 5) was present.

	Volume [%]	
	All Slices	Only 1% of Pathology
Class 1	93.72%	85.39%
Class 2	3.49%	7.7%
Class 3	0.32%	0.77%
Class 4	0.47%	1.16%
Class 5	2.01%	4.59%

**Table 2 cancers-14-01341-t002:** (**a**) Global Dice (in %) of the networks evaluated on validation subset of the stage one manual draft segmentations. (**b**) Global Dice (in %) of the stage two networks evaluated on training and validation datasets. In bold the best results are shown for a class in an individual subset of data. (weighted cross entropy loss—WCE, Lovasz loss—LV, 32 filters at first layer—32, 64 filters at first layer—64).

(a)	Stage One—Global Dice [%]							
	**Training Subset**				**Validation Subset**			
**Tissue Class**	**WCE 32**	**LV 32**	**LV 64**	**Ensemble**	**WCE 32**	**LV 32**	**LV 64**	**Ensemble**
Class 1	**99.15**	99.06	97.5	98.97	98.32	**98.45**	97.14	98.32
Class 2	**71.43**	70.23	42.85	68.78	**51.46**	48.20	32.8	48.05
Class 3	46.09	52.43	16.33	**53.37**	3.15	**14.78**	9.22	13.1
Class 4	70.47	76.53	68.45	**76.99**	41.85	44.88	43.83	**46.58**
Class 5	89.78	91.62	90.23	**91.7**	93.16	93.34	93.00	**93.69**
AVG	75.38	**77.97**	63.07	77.96	57.59	59.93	55.2	**59.95**
**(b)**	**Stage Two—Global Dice [%]**							
	**Training Subset**				**Validation Subset**			
**Tissue Class**	**WCE 32**	**LV 32**	**LV 64**	**Ensemble**	**WCE 32**	**LV 32**	**LV 64**	**Ensemble**
Class 1	96.28	98.71	**98.84**	98.81	96.65	98.98	**98.94**	98.91
Class 2	45.01	66.03	**69.41**	67.28	46.9	69.54	**72.19**	69.89
Class 3	35.13	37.46	31.8	**39.99**	27.12	32.15	26.67	**32.91**
Class 4	71.66	**80.98**	79.57	80.00	60.22	74.06	72.37	**74.31**
Class 5	89.22	**93.71**	93.36	93.57	92.86	96.21	96.22	**96.35**
AVG	67.46	75.38	74.6	**75.93**	64.75	74.19	73.28	**74.47**

**Table 3 cancers-14-01341-t003:** Comparison in percentage of segmentation (manual or automatic) scans chosen to be refined in the second stage of ground truth data generation. The segmentations chosen to be refined were closer to the target ground truth, therefore requiring fewer corrections to be made.

	Training Dataset	Validation Dataset
Manual	12.86%	18.33%
Automatic	87.14%	81.67%

**Table 4 cancers-14-01341-t004:** Global Dice (in %): performance of the individual networks and ensembles of networks on development dataset from 5-fold cross validation. Only results from validation folds are reported. The results in the green fields show scores for the network ensembles over loss functions for a particular filter number (three networks), whereas in the red fields are results for networks with different number of filters combined for a particular loss function (two networks). The violet fields show the final ensemble where all six networks were combined. The best overall global Dice scores for each class across all configurations (single networks or ensembles) are shown in bold.

		Global Dice [%]			
**Nr of Filters**	**Tissue Class**	**WCE**	**DC**	**LV**	**Ensemble**
	Class 1	94.54	98.63	97.39	98.62
	Class 2	32.88	63.69	50.72	65.42
**32**	Class 3	21.3	40.3	20.04	39.87
	Class 4	62.88	70.89	65.46	72.11
	Class 5	88.53	93.83	93.05	94.12
	Class 1	95.2	**98.74**	98.39	98.64
	Class 2	35.26	64.87	62.18	64.27
**64**	Class 3	24.63	38.19	21.82	37.55
	Class 4	61.92	71.31	73.08	70.39
	Class 5	88.26	93.77	94.11	93.8
	Class 1	95.43	**98.74**	98.51	98.7
	Class 2	36.91	65.75	64.19	**66.45**
**Ensemble**	Class 3	27.13	40.71	23.99	**40.89**
	Class 4	65.22	72.25	**73.88**	73.27
	Class 5	89.6	94.22	**94.46**	94.31

**Table 5 cancers-14-01341-t005:** We present details of the 5-fold cross validation experiment, with the mean, std, and lower and higher confidence intervals ($ci95_{lo},ci95_{hi}$) corresponding to Figure 3.

		Global Dice [%]			
**Class**	**Data Type**	* **Mean** *	* **std** *	* **ci95** * ${}_{\mathrm{hi}}$	* **ci95** * ${}_{\mathrm{lo}}$
Class 1	Training	99.06	0.13	99.12	99.00
	Validation	98.86	0.18	99.02	98.70
Class 2	Training	73.56	3.38	75.04	72.08
	Validation	68.02	2.14	69.89	66.14
Class 3	Training	54.59	7.55	57.89	51.28
	Validation	38.86	6.37	44.44	33.28
Class 4	Training	80.75	4.48	82.71	78.79
	Validation	72.03	5.92	77.22	66.84
Class 5	Training	94.31	3.23	95.73	92.90
	Validation	92.59	4.48	96.51	88.66

**Table 6 cancers-14-01341-t006:** Global Dice (in %) for ensemble of six networks on final data.

	Global Dice [%]	
**Tissue Class**	**Training**	**Testing**
Class 1	98.88	98.46
Class 2	71.38	43.71
Class 3	55.79	26.09
Class 4	79.49	47.21
Class 5	95.70	92.44

**Table 7 cancers-14-01341-t007:** A class by class qualitative evaluation of the manual and automatic segmentations by an independent observer on a case basis.

	Manual			Automated			
**Tissue Class**	**1**	**2**	**3**	**1**	**2**	**3**	
Class 1	83.3%	16.6%	0%	80%	6.7%	13.3%	1—Acceptable
Class 2	90%	10%	0%	83.3%	3.3%	13.3%	2—Minor Disagreement
Class 3	100%	0%	0%	96.7%	3.3%	0%	3—Major Disagreement
Class 4	96.7%	3.3%	0%	93.3%	6.7%	0%	
Class 5	93.3%	0%	6.7%	86.7%	3.3%	10%	
AVG All Classes	92.7%	6%	1.3%	88%	4.7%	7.3%	

## Data Availability

The data presented in this study are available on request from the corresponding author.

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
