# Peer review of "A Novel and Automated Approach to Classify Radiation Induced Lung Tissue Damage on CT Scans"

_cancers, 2022, doi:10.3390/cancers14051341_

Round 1

Reviewer 1 Report

  • The study is about the usage of CT scans using CNN. The study is novel with a great goal; however, I would like to introduce some comments to increase the impact of the paper.
  • In the introduction section: remove the enumerations. No need for 1.1. and 1.2. just separate the paragraphs.
  • A crucial point is the data labeling: As the authors did labeling, I highly recommend double-checking their annotation with an external radiologist.
  • The authors must indicate the type of measurement in Table 1: I mean 99.15 is a percentage or a number: if it was a percentage, it should be 99.15%. Repeat for all the tables.
  • Remove the pie chart, use a proper table.
  • Type the confidence interval of the k-fold validation of each calculation and increase the K to be 10.
  • Where is the network used? The authors must have a separate section called Convolution neural network and formally define CNN to define their novel architecture. The authors should also include a diagram of their network.
  • Why did the authors not compare their network to other SoTA networks? They must compare it to others to emphasize their importance of theirs.
  • Where are the hyperparameters used in this study? The authors should make a table with all the hyperparameters used.

Author Response

We thank the reviewer for their comments. We attached a document where we addressed them all.

Reviewer 2 Report

The work is devoted to radiation-induced lung damage (RILD). The authors propose a novel classification scheme which is, contrary to existing methods, based on morphological, and not on pathophysiological features. The authors also apply UNet CNN to automate segmentation. To improve quality of manually-labeled masks, they apply an active learning approach using the same CNN to generate possibly improved masks, and manually select the best masks. They use 5-fold cross-validation approach to evaluate the training on different subsets. Finally, they evaluate the segmentation results on the test set. The prediction results are not always impressive, but considering the dataset size and diversity of imaging parameters, in my opinion they are acceptable (the paper seems to be more proof of concept). The authors also analyze reasons of misclassification.

I have several major and minor comments.

Major comments:

  1. There some missing training parameters which could be interesting. Which were number of epochs, learning rate, type of optimizer, patience, etc.? Stop criteria? Initial image size? Did you try to tune these parameters?
  2. Similar question about the last layer. Was this sigmoid activation function and class-per-class segmentation or multiclass segmentation with softmax activation function? Why this was the best choice?
  3. Which was the post-processing (binary thresholding with 0.5 cutoff probability?)? Did you try other types of post-processing?
  4. Line 243, there is class imbalance which is mentioned. Did you try to introduce class weights?
  5. The text could be shortened. For example, mentioning that only images with >1% pathological tissues were used for the training, is repeated in lines 242, 271, 397, 405, Figure 1, Table2. At least a half of these could be removed
  6. In my opinion the “speculation” part of the conclusion should be removed from the conclusions, and some more results should be added.

Minor comments:

  1. I have some suggestion about the general structure. Subsection 2.1 could be renamed (imaging parameters are not materials, but methods). Order of 2.3 and 2.4 could be changed, maybe subsection 2.3 could be placed to 2.4.1.
  2. Lines 20-21, maybe it would be better to mention that “from normal lung to consolidation” corresponds to Class 1 to Class 5, otherwise it is difficult to understand without reading the main text.
  3. Lina 189 mm in italic font.
  4. Line 279 “we decided” sounds not scientifically, it would be better to underline that this was a hypothesis.
  5. Section 3.3 the title is not precise: not the test dataset was evaluated, but the segmentation quality.

Author Response

(The authors gave the same response as above.)

Reviewer 3 Report

The application value is obvious, but the academic value is not high.
(1)26%~98% in Simple Summary. 26% seems too low.
(2)The contributions are too simple at the end of Section 1. Why are they contributions? What is are originality and difficulty for each point?
(3)Fig. 2 is too simple. The authors should plot the framework of the networks and all the details of the models, such as the parameters.
(4)According to Section 2.4.1, is “ensemble” simply the summing of the scores? It is better to analyze the correlations among the scores. (Ref. PalmHash Code vs. PalmPhasor Code can be cited and helpful.)
(5)There is no ablation experiment. The comparison experiments are insufficient. Some SOTA segmentation models are helpful. (E.g. -Mask Refined R-CNN: A network for refining object details in instance segmentation) If the authors cannot employ these methods or compare their method with these methods, at least they could introduce/mention these novel technologies to improve the quality of the survey, or explain them as the possible future works.
(6)Some paragraphs are too long and difficult to follow, e.g. the paragraphs in Section 4. Please divide them into several short paragraphs to improve the readability. It is better if the authors could state the discussions point-by-point.
(7)Pages 22 and 23 are empty and should be deleted.

Author Response

(The authors gave the same response as above.)

Round 2

Reviewer 2 Report

Thank you for the corrections, all my suggestions were taken into account.

Reviewer 3 Report

Accept.